# Exosomal microRNA let-7-5p from *Taenia pisiformis* Cysticercus Prompted Macrophage to M2 Polarization through Inhibiting the Expression of C/EBP δ

**DOI:** 10.3390/microorganisms9071403

**Published:** 2021-06-29

**Authors:** Liqun Wang, Tingli Liu, Guoliang Chen, Yanping Li, Shaohua Zhang, Li Mao, Panhong Liang, Majid Fasihi Harandi, Taoshan Li, Xuenong Luo

**Affiliations:** 1State Key Laboratory of Veterinary Etiological Biology, Key Laboratory of Veterinary Parasitology of Gansu Province, Lanzhou Veterinary Research Institute, Chinese Academy of Agricultural Sciences (CAAS), Lanzhou 730046, China; wlq1282690114@163.com (L.W.); LTL1114@163.com (T.L.); glchen2019@163.com (G.C.); lyyp223@163.com (Y.L.); zhangshaohua01@caas.cn (S.Z.); mao-li@live.cn (L.M.); liangpanhong628@163.com (P.L.); 18394490562@163.com (T.L.); 2Research Center for Hydatid Disease in Iran, Kerman University of Medical Sciences, Kerman 7616914115, Iran; majid.fasihi@gmail.com; 3Jiangsu Co-Innovation Center for the Prevention and Control of Important Animal Infectious Disease and Zoonoses, Yangzhou University, Yangzhou 225009, China

**Keywords:** *Taenia pisiformis* cysticercus, exosome, miRNA let-7-5p, macrophages polarization

## Abstract

*Cysticercus pisiformis*, the larval stage of *Taenia pisiformis*, causes serious illness in rabbits that severely impacts the rabbit breeding industry. An inhibitive Th2 immune response can be induced by let-7-enriched exosomes derived from *T. pisiformis* cysticercus. However, the underlying molecular mechanisms are not completely understood. Here, we report that exosomal miR-let-7-5p released by *T. pisiformis* cysticercus played a critical role in the activation of M2 macrophages. We found that overexpression of let-7-5p in M1 macrophages decreased M1 phenotype expression while promoting polarization to the M2 phenotype, which is consistent with experimental data in exosome-treated macrophages alone. In contrast, knockdown of let-7-5p in exosome-like vesicles promoted M1 polarization and decreased M2 phenotype expression. Furthermore, down-regulation of transcription factor CCAAT/enhancer-binding protein (C/EBP)-δ resulted in the decrease of M1 phenotype markers and increase of M2 phenotype markers. These results suggested that let-7 enriched in exosome-like vesicles from *T. pisiformis* metacestodes can induce M2 macrophage polarization via targeting C/EBP δ, which may be involved in macrophage polarization induced by *T. pisiformis* metacestodes. The finding helps to expand our knowledge of the molecular mechanism of immunosuppression and Th2 immune response induced by metacestodes.

## 1. Introduction

Parasitic diseases remain a serious global health issue, threatening the lives of a third of the global population. Humans and animals are involved in certain stages of parasite life cycles [1]. The larval stage of *Taenia pisiformis* (*T. pisiformis*), also known as *T. pisiformis* metacestode, is one of the most common parasites in farmed and wild rabbits [2]. *T. pisiformis* metacestode infection causes significant health problems in lagomorphs, such as weight loss, reduction of litter size [3,4,5] and death, leading to considerable economic losses in the rabbit breeding industry.

During the long-term coevolution between helminths and hosts, helminths have developed various mechanisms that regulate and evade the host immune system, ensuring their persistence within the host [6], that are partially attributed to the secretion of molecules with immunomodulatory properties. Although the excretory/secretory molecules of helminths have been extensively studied [7,8], emerging studies highlight the importance of helminth extracellular vesicles (EVs) and their miRNAs in host-parasite interactions [9,10]. For example, exosomes derived from *Schistosoma japonicum* induce M1-type macrophage polarization in vitro [11]. miR-71-enriched EVs can inhibit the production of nitric oxide (NO) through downregulation of iNOS in RAW264.7 macrophages [12]. In the gastrointestinal nematode *Heligmosomoides polygyrus*, Buck et al. [13] identified secreted exosomes that contained miRNAs and Y RNA that suppressed the expression of Il33r and Dusp1, suggesting exosomal miRNAs could regulate target gene expression in host cells. Although our previous results showed that Th2 immune response inhibition can be induced by exosome-like vesicles derived from *T. pisiformis* metacestodes both in immunized rabbits in vivo and macrophages in vitro [14], the exact molecular mechanisms regarding the exosome-like vesicles in modulating of Th2 immune response/M2 macrophage polarization requires further investigation.

MicroRNAs (miRNAs) are a class of conserved non-coding RNAs that bind to complementary sites in the 3′ untranslated region (3′ UTR) of target mRNAs and lead to either translation inhibition or mRNA degradation [15,16]. The let-7 miRNA family is involved in developmental transitions, cell signaling, and the regulation of immune response [17,18]. In addition, vertebrate miRNA let-7c can polarize macrophages toward the M2 phenotype by regulating the expression of C/EBP-δ [19], and our previous study demonstrated miRNA let-7 is one of the most abundant miRNAs in exosome-like vesicles of *T. pisiformis* metacestodes [14]. Thus, we speculated that *T. pisiformis* metacesode exosome-like vesicles were also involved in macrophage polarization through exosomal let-7 mediated gene regulation, thereby inhibiting the host inflammatory response. To explore the role of exosome miRNA let-7 in regulating macrophage polarization, we isolated the exosomes from *T. pisiformis* using ultracentrifugation and investigated the effect of overexpression or knockdown of exosome let-7-5p of *T. pisiformis* on exosome-induced M2 macrophage polarization. The results showed that *T. pisiformis*-derived exosome delivers miRNA let-7-5p to induce M2 macrophage polarization by targeting C/EBP δ. The findings will assist clarification of the mechanisms of macrophage polarization and immune evasion induced by *T. pisiformis* metacestode-derived exosomes from a new perspective and will facilitate advanced investigations into anti-inflammatory drugs for the treatment of cysticercosis.

## 2. Materials and Methods

### 2.1. Ethics Statement

All animal experiments were approved by the Animal Administration and Ethics Committee of Lanzhou Veterinary Research Institute, Chinese Academy of Agricultural Sciences (Permit No. LVRIAEC-2016-006). Animal experiments were performed in strict compliance with the recommendations in the Guide for the Care and Use of Laboratory Animals of the Ministry of Science and Technology of the People’s Republic of China.

### 2.2. T. pisiformis Larvae Cultures and Exosome-Like Vesicles Isolation

All experimental animals were purchased from the Laboratory Animal Center of Lanzhou Veterinary Research Institute. New Zealand white rabbits were orally challenged with 500 *T. pisiformis* eggs. *T. pisiformis* larvae were collected from the peritoneal cavities of rabbits at 50 days post-infection and washed with sterile PBS supplemented with 100 units/mL penicillin and 100 μg/mL streptomycin (Life Technologies Inc., Grand Island, NY, USA). The metacestodes were maintained in preheated RPMI-1640 medium (Gibco, Grand Island, NY, USA) containing 10% exosome-depleted fetal bovine serum (FBS), 100 μg /mL streptomycin and 100 units/mL penicillin in a training environment at 37 °C and 5% CO_2_. Culture media were refreshed after 12 h to remove contaminants from host components. The excretory/secretory products (ESPs) of *T. pisiformis* cysticerci were collected at 24 h and 48 h. Isolation of exosome-like vesicles was carried out from pooled ESPs as described in our previous study [14]. The size of isolated exosome-like vesicles was determined using a NanoSight LM10 instrument (Nanosight, Wiltshire, UK). The vesicle protein concentration was determined using a Pierce BCA Protein Assay Kit (Thermo Fisher Scientific, Waltham, MA, USA).

### 2.3. Transmission Electron Microscopy (TEM) and Immune TEM (ITEM)

Transmission electron microscopy (TEM, Hitachi Ltd., Tokyo, Japan) was employed to observe the size and morphology of the exosome-like vesicles. The purified exosome-like vesicles (10 μL) were placed into a 200-mesh formvar-coated copper grid (Agar Scientific Ltd., Stansted, UK) and negatively stained with phosphotungstic acid solution (3%, pH 7.0) for 1 min. Furthermore, ITEM for exosome-like vesicles was carried out as previously described [20]. Briefly, the purified exosome-like vesicles were resuspended with an equal volume of 4% (*w*/*v*) paraformaldehyde and deposited on 200 mesh formvar-carbon coated grids for 20 min. The grids were washed with 50 mM glycine and blocked with 5% bovine serum albumin (BSA) (Sigma-Aldrich, St. Louis, MI, USA). The vesicles were incubated with the primary antibody for 20 min, which included polyclonal rabbit anti-CD63 (1:200, Abcam, Cambridge, MA, USA) and polyclonal mouse anti-CD9 (1:200, Abcam, Cambridge, MA, USA). Subsequently, the grids were washed six times with PBS and labeled with goat-anti-rabbit/goat-anti-mouse IgG coupled to 10 nm gold particles (Jackson Immunoresearch, West Grove, PA, USA) for 45 min at room temperature (RT). The samples were subsequently washed eight times with distilled water and stained with 3% phosphotungstic acid (Sigma-Aldrich, St. Louis, MI, USA) in the dark for 1 min. The vesicles were observed under the Hitachi electron microscope at 80 kV.

### 2.4. Absolute Quantification of let-7-5p in Exosome-Like Vesicles

To quantify the level of let-7-5p in the exosome-like vesicles and cysticercus of *T. pisiformis*, a miRNA let-7-5p standard sample (10 μM) was synthesized by Guangzhou RiboBio Co., Ltd. (Guangzhou, China). Absolute quantification of let-7-5p was performed using quantitative real-time polymerase chain reaction (qPCR) according to the manufacturer’s instructions as previously described [21]. Briefly, 10-fold serial dilutions (1 × 10^7^ to 1 × 10^2^ copies/μL) of the cDNA products derived from synthetic let-7-5p oligonucleotides were used to construct the standard curve and each dilution was repeated three times. Prior to RNA extraction, *Caenorhabditis elegans* miRNA cel-miR-39-3p was added to each sample as an external control to monitor miRNA extraction efficiency. 10 μg of exosome-like vesicles and 50 mg of *T. pisiformis* cysticercus were used for total RNA extraction using TRIzol (Thermo Fisher Scientific, Waltham, MA, USA), according to the manufacturer’s instructions. Using the RT primers (Table 1), first-strand cDNA of miRNAs was synthesized using 1 μg of total RNA using a Mir-X™ miRNA First-Strand Synthesis Kit (Takara, Dalian, China) according to the manufacturer’s protocols. qPCR was performed on an ABI 7500 using a TransStart Tip Green qPCR SuperMix Kit (TransGen Co., Beijing, China) with the following protocol: denaturation at 95 °C for 30 s, followed by 40 cycles of 95 °C for 15 s and 60 °C for 34 s. The copy number of let-7-5p was calculated according to the standard curve.

### 2.5. Cell Culture and Macrophage Polarization

RAW264.7 macrophage cells were provided by the Stem Cell Bank, Chinese Academy of Sciences (Shanghai, China) and maintained in DMEM (Gibco, NY, USA) supplemented with 10% FBS (Gibco). The macrophages were polarized to an M1 phenotype with 1000 ng/mL LPS and 10 ng/mL IFN-γ as previously described [22].

### 2.6. Exosome-Like Vesicles Treatment and miRNA Mimics/Inhibitor Transfection

M1 macrophages were seeded in 6-well plates at 1 × 10^5^ cells per well. After 24 h, the cells were treated with 50 μg/mL exosome-like vesicles of *T. pisiformis* cysticercus or PBS for 24 h. Based on the let-7-5p copy number in the exosome-like vesicles, 50 nM let-7-5p mimic and 50 nM control negative mimic (RiboBio Co., Ltd., Guangzhou, China) equivalent to let-7-5p in 50 μg/mL exosome-like vesicles were transfected into M1 macrophages using Lipofectamine™ RNAiMAX Transfection Reagent (Thermo Fisher Scientific, Waltham, MA, USA) according to the manufacturer’s protocol. Additionally, to downregulate the expression of miRNA let-7-5p, exosome-treated M1 macrophages were transfected with 50 nM control negative inhibitor and 50 nM let-7-5p inhibitor (RiboBio Co., Ltd., Guangzhou, China) having the same copy number of 50 μg/mL exosmal let-7-5p.

### 2.7. qPCR Analysis of the M1/M2 Marker Relative Expression

Total RNA from RAW264.7 cells were extracted using TRIzol reagent according to the manufacturer’s instructions. First-strand cDNA synthesis was conducted using 1 μg of total RNA with HiScript II 1st Strand cDNA Synthesis Kit (Vazyme, Nanjing, China). qPCR was performed on an ABI 7500 Thermal Cycler (Thermo Fisher Scientific, Waltham, MA, USA) using All-in-One qPCR Mix (Genecopoeia, Guangzhou, China) under the following conditions: 95 °C for 10 min, 40 cycles for two steps (95 °C for 5 s and 60 °C for 1 min). The murine glyceraldehyde-3-phosphate dehydrogenase (GAPDH) was used as the internal reference gene. The 2^−ΔΔCq^ method was used to calculate relative mRNA abundance [23]. The qPCR primers used in this study are listed in Table 1.

### 2.8. Sequence Conservation and Target Genes Analysis of let-7-5p

To evaluate the sequence conservation of let-7-5p, the mature sequences from 19 different species were collected from the miRBase database. The multi-sequence alignment software ClustalX (version 2.1) (ftp://ftp.ebi.ac.uk/pub/software/clustalw2/ (accessed on 12 July 2019)) was used for the alignment analysis of let-7 among the different species. The target genes of let-7-5p were predicted using RNAhybrid (http://bibiserv.techfak.uni-bielefeld.de/rnahybrid/ (accessed on 15 October 2019)), miRanda (http://www.microrna.org/microrna/home.do/ (accessed on 15 October 2019)) and TargetScan software (http://www.targetscan.org/ (accessed on 15 October 2019)).

### 2.9. Luciferase Assay

The sequence of the transcription factor C/EBP δ containing the wild-type or mutated binding site of let-7-5p was synthesized and cloned into pmirGLO vectors (Promega, Madison, WI, USA). These recombinant plasmids were co-transfected into HEK-293T cells with let-7-5p mimics, negative control mimics, let-7-5p inhibitor or negative control inhibitor. 24 h after transfection, relative luciferase activity of the transfected cells was measured using the Dual Glo Luciferase assay system (Promega, WI, USA) according to the manufacturer’s instructions.

### 2.10. Western Blot

RAW264.7 macrophage cell samples from the different groups were collected, washed three times with PBS and lysed in RIPA lysis buffer (Beyotime, Shanghai, China) containing the halt protease inhibitor cocktail (100× stock diluted to 2×) (Thermo Fisher Scientific, Waltham, MA, USA). The protein concentration of the lysates was measured using a BCA protein assay kit. 15 μg of lysates were separated by 12% SDS-PAGE and electrophoretically transferred to polyvinylidene difluoride (PVDF) membranes (Millipore, Bedford, MA, USA). After three washes with PBS, the membranes were blocked with 5% fat-free milk in PBST for 2 h at RT, then incubated with primary antibodies against C/EBP δ (1:1000 dilution in PBST) (Abcam, Cambridge, MA) at 4 °C overnight. The membranes were washed thrice with PBST and incubated with horseradish peroxidase (HRP)-conjugated secondary anti-mouse antibody for 1 h at RT. The membranes were washed with PBST three times, and the bands were detected using ECL chemiluminescence working solution (Beyotime, Shanghai, China), following the manufacturer’s instructions.

### 2.11. Statistical Analysis

Data were analyzed using SPSS version 21.0 software (SPSS Inc, Chicago, IL, USA). Comparisons between different groups were conducted using one-way ANOVA/Student’s t-tests. *p* < 0.05 was considered statistically significant.

## 3. Results

### 3.1. Exosome-Like Vesicles Secreted by T. pisiformis Metacestodes

Using a protocol for *T. pisiformis* metacestodes exosme-like vesicle isolation we developed previously, exosome-like vesicles were isolated from serum-free ESP of *T. pisiformis* cysticercus. Purified exosome-like vesicles were visualized by transmission electron microscopy (TEM), which revealed the presence of spherical vesicles (Figure 1A). Further, ITEM analysis indicated that the isolated vesicles contained two exosomal markers CD9 and CD63 (Figure 1B). The size distribution of the particles was shown to range predominantly from 50 nm to 200 nm in diameter (Figure 1C), consistent with our previous study [14]. Taken together, the TEM, ITEM and NTA analyses confirmed that exosome-like vesicles were successfully isolated from ESPs of *T. pisiformis* cysticercus.

### 3.2. Exosomal miRNA let-7-5p Derived from T. pisiformis Cysticercus Promoted M2 Macrophage Activation

To determine the expression level of miR-let-7-5p, we performed an absolute quantification for miR-let-7-5p in exosome-like vesicles and cysticercus of *T. pisiformis* (Figure 2). The result showed that the copy number of miR-let-7-5p in exosome-like vesicles and metacestodes of *T. pisiformis* were 2.25 × 10^6^ per 1 μg total RNA and 2.74 × 10^6^ per 1 μg total RNA, respectively. To evaluate the effect of exosome-like vesicles from *T. pisiformis* cysticercus on macrophage polarization, M1 phenotype macrophages were treated with exosome-like vesicles. The qPCR analysis showed that, compared to the M1+PBS group, the vesicle treatment significantly attenuated LPS and IFN-γ induced expression of M1 macrophage marker genes IL-12 and iNOS (Figure 3A,B), while markedly increasing the production of M2 macrophage marker genes IL-10 and Arg-1 (Figure 3C,D). These results showed that exosome-like vesicles from *T. pisiformis* cysticercus could induce M2 macrophage activation. The miR-let-7-5p expression in macrophages was measured and the qPCR results demonstrated that miR-let-7-5p expression increased significantly in a dose-dependent manner after *T. pisiformis* cysticercus derived exosome treatment (*p* < 0.05) (Figure 4). To determine whether miR-let-7-5p was the key component inducing M2 macrophage activation, we examined the effect of overexpression of miR-let-7-5p on macrophages polarization. The results showed that miR-let-7-5p significantly induced M2 macrophage polarization, indicated by up-regulation of M2 macrophage markers Arg-1 and IL-10 and down-regulation of M1 macrophage markers iNOS and IL-12 (Figure 3). Moreover, knockdown of miR-let-7-5p expression by transfection of miR-let-7-5p inhibitor in M1 macrophages significantly decreased M2 macrophage polarization (Figure 3).

### 3.3. C/EBP δ Was a Direct Target of miR-let-7-5p

To identify potential targets of miR-let-7-5p in promoting M2 macrophage polarization, we searched target genes of miR-let-7-5p that could participate in this process and found that transcription factor C/EBP δ is one of the target mRNAs of miR-let-7-5p. C/EBP δ has been reported to play an important role in the induction of inflammatory cytokines [24].

The multi-sequence alignment results (Figure 5) showed that the seed sequence (nucleotide 2 to 8) of let-7 was completely identical in 19 different species, indicating that let-7 are highly conserved among different species. The sequence alignment in Figure 6A shows that miR-let-7-5p has a binding site in the 3′ UTR of wild-type C/EBP δ. To verify whether C/EBP δ is a target gene of miR-let-7-5p, luciferase reporter plasmids containing wild type (WT) or mutant (Mut) C/EBP δ 3′ UTR were constructed and co-transfected with miR-let-7-5p mimics or miR-let-7-5p inhibitor into HEK293T cells. Transfection of miR-let-7-5p mimics markedly inhibited luciferase activity of C/EBP δ 3′ UTR-WT, while knockdown of miR-let-7-5p significantly increased luciferase activity. Neither miR-let-7-5p mimics nor inhibitor had an effect on C/EBP δ 3′ UTR-Mut (Figure 6B). In addition, qPCR and Western blot analysis showed that overexpression of miR-let-7-5p significantly inhibited the expression of C/EBP δ, while knockdown of miR-let-7-5p significantly reduced C/EBP δ expression in macrophages (*p* < 0.05; Figure 6C,D), suggesting that C/EBP δ is a direct target gene of miR-let-7-5p.

### 3.4. C/EBP δ Was Involved in Macrophage Polarization

To determine whether C/EBP δ is related to macrophage polarization, we examined the expression of M1/M2 macrophages markers in M1 macrophages that were transfected with specific C/EBP δ small interfering RNA (siRNA) or control siRNA. As shown in Figure 6, knockdown of C/EBP δ (Figure 6E) significantly decreased the mRNA levels of M1 phenotype marker molecules IL-12 and iNOS (Figure 6F,G). In contrast, C/EBP δ knockdown significantly enhanced the expression of M2 phenotype, as indicated by increased levels of IL-10 and Arg-1 (Figure 6H,I). These data suggest that knockdown of C/EBP δ inhibited M1 polarization and promoted M2 polarization and were consistent with the results of miR-let-7-5p overexpression, suggesting that C/EBP δ is involved in the regulation of M2 macrophage polarization.

## 4. Discussion

In this study, we isolated exosome-like vesicles from *T. pisiformis* larvae based on the protocol developed in our previous study. TEM and NTA analysis identified vesicles with circular or elliptical vesicular structures ranging from 50 to 200 nm in diameter, the shape and size of which were similar to those described in our previous study and to those released by other helminths [14], such as *Taenia asiatica*, *Echinococcus granulosus*, and *Echinococcus multilocularis* [12,20,25]. The transmembrane proteins CD9 and CD63 are considered as exosomal markers and are frequently used to identify exosomes after isolation [26,27]. Our ITEM analysis results showed that CD9 and CD63 were detected in exosome-like vesicles from *T. pisiformis* metacestodes. Collectively, the results confirmed that high-quality *T. pisiformis* metacestodes exosomes were obtained and that the isolated exosome-like vesicles could be used in subsequent experiments.

MiRNAs are a class of small non-coding RNAs that regulate the expression of target genes [28]. In vertebrates, the miRNA let-7 family is composed of 12 genes members, encoding 9 different miRNAs (let-7a, let-7b, let-7c, let-7d, let-7e, let-7f, let-7g, let-7h, let-7i and miR-98), and their targets are involved in cell proliferation, apoptosis and innate immunity [29]. Studies have shown that let-7c plays an important role in macrophage polarization [30]. Metacestodes and their exosomes contain only a highly abundant let-7-5p, similar to the vertebrate let-7 [31], but whether the let-7-5p derived from exosomes of *T. pisiformis* metacestodes has the same function as vertebrate let-7c in modulating macrophage polarization remained unclear. The present study found that exosome and let-7-5p from *T. pisiformis* metacestodes promoted macrophages polarization to the M2 phenotype. Multiple evidence in our study supports that target gene C/EBP δ is regulated by let-7-5p to regulate macrophage phenotype. First, the dual luciferase reporter assay confirmed that the 3′ UTR of C/EBP δ contains a let-7-5p complementary binding site. Secondly, overexpression of let-7-5p significantly down-regulated the expression of the C/EBP δ in RAW264.7 macrophages at both at mRNA and protein levels, while knockdown of let- 7-5p significantly up-regulated C/EBP δ expression. Finally, knockdown of C/EBP δ diminished M1 macrophage activation while enhancing M2 polarization. In summary, exosomal let-7-5p can regulate macrophage polarization by targeting C/EBP δ.

Recent studies have shown that transcription factors play important roles in the inflammatory response, and that mammalian let-7 induces macrophages to polarize to the M2 phenotype by regulating the expression of transcription factors [19]. It has been demonstrated that exosomes derived from LPS-preconditioned mesenchymal stromal cells upregulate the expression of anti-inflammatory cytokines and promote M2 macrophage polarization by shuttling let-7b, which can alleviate inflammation and enhance diabetic cutaneous wound healing [32]. Furthermore, exosomes from adipose-derived mesenchymal stem cells improve the survival of fat grafts by regulating macrophage polarization via let-7c [33]. Consistent with these studies, the present study demonstrated that miR-let-7-5p derived from exosome-like vesicles of *T. pisiformis* metacestodes promotes the polarization of macrophages to the M2 phenotype through inhibiting the expression of transcription factor C/EBP δ. The immune system shifts mainly toward a Th2 polarized immune response in metacestode infection [34,35]. From this we speculate that after infection, let-7 from *T. pisiformis* metacestodes can be transported into host macrophages via exosomes to regulate the polarization of macrophages to the M2 phenotype, thereby producing immunosuppression and a Th2 type cell immune response, which is beneficial to survival of larvae in the host.

In conclusion, miRNA let-7 enriched in exosome-like vesicles from *T. pisiformis* metacestodes can induce M2 macrophage polarization through targeting transcriptional factor C/EBP δ, which may be involved in macrophage polarization induced by *T. pisiformis* metacestodes. These findings help to expand our knowledge of the molecular mechanism of immunosuppression and Th2 type immune response induced by metacestodes and provide novel avenues for the research and development of new anti-inflammatory drugs for cysticercosis.

## Figures and Tables

**Figure 1 microorganisms-09-01403-f001:**
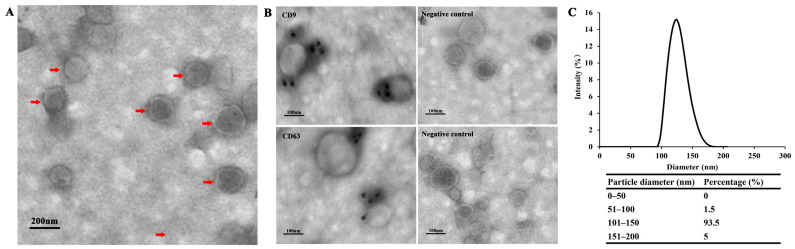
Characterization of exosome-like vesicles isolated from the culture medium of *T. pisiformis* cysticercus. (**A**) Morphological characterization of *T. pisiformis* cysticercus exosome-like vesicles by TEM. The red arrowheads indicate the exosome-like vesicles negatively stained with phosphotungstic acid. (**B**) Immunoelectron microscopy analysis of CD9 and CD63 proteins in the purified exosome-like vesicles. (**C**) Size distribution of exosome-like vesicles of *T. pisiformis* cysticercus by NTA.

**Figure 2 microorganisms-09-01403-f002:**
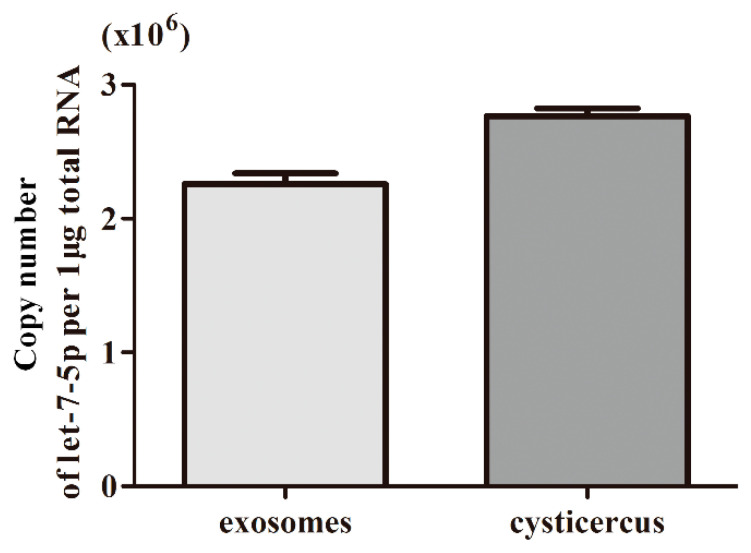
The copy number of miR-let-7-5p in exosome-like vesicles and cysticercus of *T. pisiformis*. Data were from 3 independent experiments.

**Figure 3 microorganisms-09-01403-f003:**
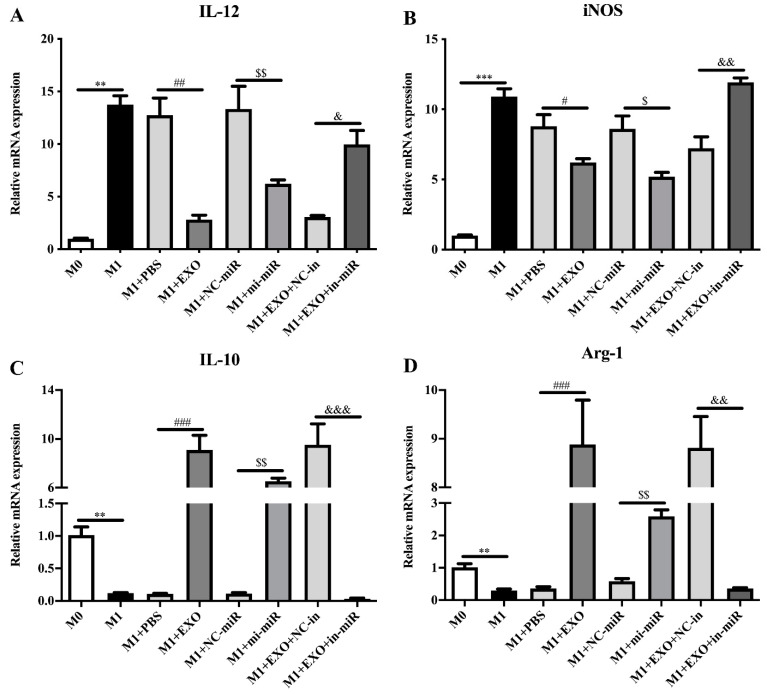
Exosomal miR-let-7-5p promoted M2 macrophage activation. (**A**) The expression levels of IL-12 after different treatments were determined by qPCR. (**B**) The expression levels of iNOS after different treatments were detected by qPCR. (**C**) The expression levels of IL-10 after different treatments were assessed by qPCR. (**D**) The expression levels of Arg-1 after different treatments were evaluated by qPCR. *n* = 3, mean ± SD, ** *p* < 0.01, *** *p* < 0.001 vs. the M0 group. # *p* < 0.05, ## *p* < 0.01, ### *p* < 0.001 vs. the M1+EXO group. $ *p* < 0.05, $$ *p* < 0.01, vs. the M1+mi-miR group. & *p* < 0.05, && *p* < 0.01, &&& *p* < 0.001 vs. the M1 + EXO + in-miR group.

**Figure 4 microorganisms-09-01403-f004:**
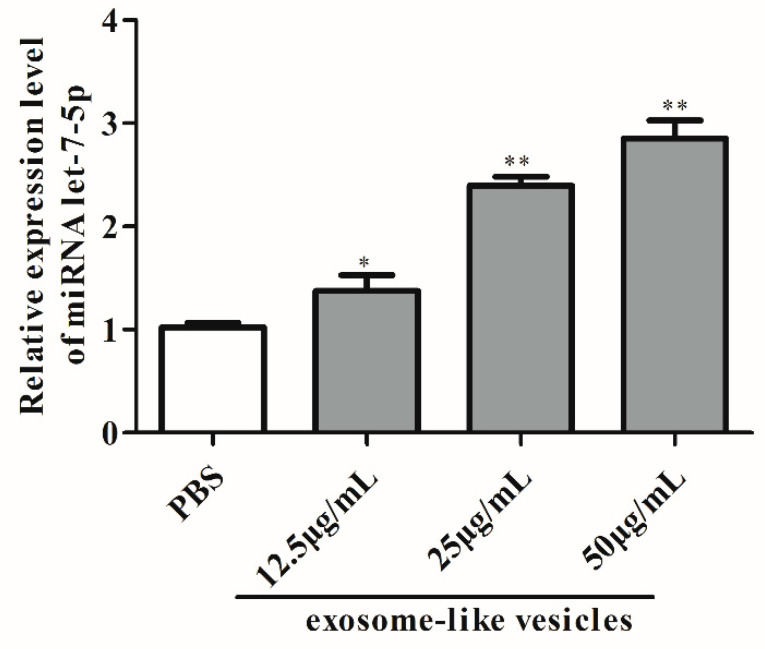
miR-let-7-5p expression in exosomes treated macrophages was detected by qPCR. *n* = 3, mean ± SD, * *p* < 0.05, ** *p* < 0.01 vs. the PBS group.

**Figure 5 microorganisms-09-01403-f005:**
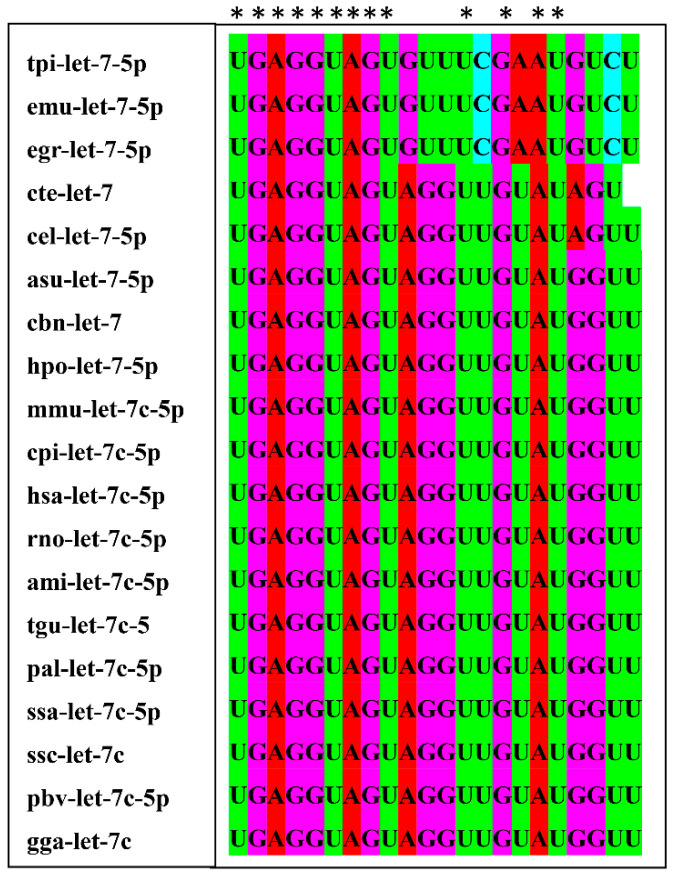
The multi-sequence alignment of let-7 among different species. Identical nucleotides are marked with *. tpi: *Taenia pisiformis*; emu: *Echinococcus multilocularis*; egr: *Echinococus granulosus*; cte: *Capitella teleta*; cel: *Caenorhabditis elgans*; asu: *Ascaris suum*; cbn: *Caenorhabditis brenneri*; hpo: *Heligmosomoides polygyrus*; mmu: *Mus musculus*; cpi: *Chrysemys picta*; has: *Homo sapiens*; rno: *Rattus norvegicus*; ami: *Alligator mississippiensis*; tgu: *Taeniopygia guttata*; pal: *Pteropus alecto*; ssa: *Salmo salar*; ssc: *Sus scrofa*; pbv: *Python bivittatus*; gga: *Gallus gallus.*

**Figure 6 microorganisms-09-01403-f006:**
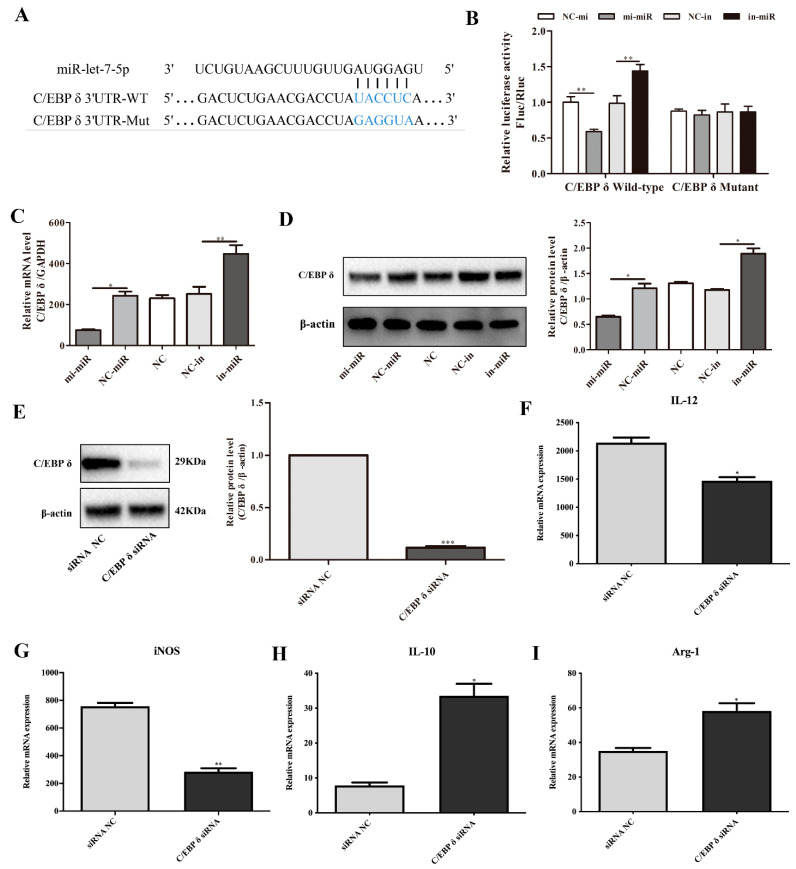
Exosomal miR-let-7-5p prompted M2 macrophage polarization via directly targeting C/EBP δ. (**A**) The predicted binding site between miR-let-7-5p and C/EBP δ 3′ UTR. (**B**)The activities of luciferase reporters containing wild-type (WT) or mutant (MUT) C/EBP δ 3′ UTR were transfected into HEK293T cells that were then treated with miR-let-7-5p mimic, inhibitor, or control. ** *p* < 0.01 vs. the control group. (**C**) mRNA expression of C/EBP δ in macrophages after miR-let-7-5p mimic or control transfection. * *p* < 0.05 vs. the control group. (**D**) Western blot analysis of C/EBP δ in macrophages after miR-let-7-5p mimic or control transfection. * *p* < 0.05 vs. the control group. (**E**) Western blot analysis of C/EBP δ in macrophages after 50 nM siRNA or 50 nM siRNA control transfection. * *p* < 0.05 vs. the control group. (**F**) Levels of IL-12 were determined by qPCR. (**G**) Levels of iNOS were detected by qPCR. (**H**) Levels of IL-10 were assessed by qPCR. (**I**) Levels of Arg-1 were evaluated by qPCR. *n* = 3, mean ± SD, * *p* < 0.05, ** *p* < 0.01, *** *p* < 0.001 vs. the control group.

**Table 1 microorganisms-09-01403-t001:** The primers used in this study.

Primer	Sequence (5′–3′)
iNOS forward primer	CGAAACGCTTCACTTCCAA
iNOS reverse primer	TGAGCCTATATTGCTGTGGCT
Arg-1 forward primer	TCATGGAAGTGAACCCAACTCTTG
Arg-1 reverse primer	TCAGTCCCTGGCTTATGGTTACC
IL-10 forward primer	GCTCTTACTGACTGGCATGAG
IL-10 reverse primer	CGCAGCTCTAGGAGCATGTG
IL-12 forward primer	TTGCTGGTGTCTCCACTCATG
IL-12 reverse primer	GTCACAGGTGAGGTTCACTGTTTC
GAPDH forward primer	GTCTTCACCACCATGGAG
GAPDH reverse primer	CCAAAGTTGTCATGGATGACC
RT primer	GCGAGCACAGAATTAATACGACTCACTATAGG(T)_12_VN ^a^
Universal reverse primer	GCGAGCACAGAATTAATACGAC
miR-let-7-5p	TGAGGTAGTGTTTCGAATGTCT

^a^ ‘V’ stands for A, G or C; ‘N’ stands for A, T, G or C.

## Data Availability

Data supporting the conclusions of this article are included within the article.

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
