# Peer review of "Exosomal microRNA let-7-5p from Taenia pisiformis Cysticercus Prompted Macrophage to M2 Polarization through Inhibiting the Expression of C/EBP δ"

_microorganisms, 2021, doi:10.3390/microorganisms9071403_

Round 1

Reviewer 1 Report

I congratulate the authors for the research quality, a well-designed investigation, structured methodology, robust bibliographic and correct scientific writing. Just a few points, I would like to comment.

I - Structure

  1. I suggest reviewing legends and italic word presentation, concerning parasite names.

II - General Considerations

  1. Introduction - should be clear the authors of some sentences. Line 70 and follows – it is speculated (by whom?)
  2. Line 72-76 – more adequate in Material and Methods (a short summary not necessary) and could be replaced for some introductory information from other authors.
  3. Material and Methods, well described, however SPSS version 17.0 is not a present version. Confirm.
  4. Results well-presented and described
  5. Discussion - isolated exosome-like vesicles from T. pisiformis larvae based on the protocol developed in our previous study. Reference?
  6. Clear and objective conclusion

In general, very interesting scientific and academic paper, enriched by small corrections or suggestions acceptance. Recommended to publish.

Author Response

Reviewer 1

I congratulate the authors for the research quality, a well-designed investigation, structured methodology, robust bibliographic and correct scientific writing. Just a few points, I would like to comment.

I – Structure

  1. I suggest reviewing legends and italic word presentation, concerning parasite names.

Response: Thank you very much for your favourable comments and constructive suggestions on our manuscript (MS). We have carefully reviewed legends and italic word presentation and revised the format of parasite names in our MS strictly according to your suggestions.

II - General Considerations

  1. Introduction should be clear the authors of some sentences. Line 70 and follows – it is speculated (by whom?)

Response: Thank you very much for constructive suggestions on our MS. We have replaced “it is speculated” as “we speculated” in line 70 and follows.

  1. Line 72-76 – more adequate in Material and Methods (a short summary not necessary) and could be replaced for some introductory information from other authors.

Response: Thank you very much for your constructive suggestions on our MS. We have revised Line 72-76 in Material and Methods as suggested.

  1. Material and Methods, well described, however SPSS version 17.0 is not a present version. Confirm.

Response: Thank you very much for your favourable comments and constructive suggestions on our MS. We confirmed that SPSS version used in our MS is SPSS version 21.0 software (SPSS Inc, Chicago, IL, USA). We have revised accordingly in Material and Methods section.

  1. Results well-presented and described

Response: Thank you very much for your favourable comments on our MS.

  1. Discussion - isolated exosome-like vesicles from pisiformis larvae based on the protocol developed in our previous study. Reference?

Response: Thanks for your constructive suggestions on our MS. We have cited the reference accordingly in first paragraph of discussion section.

  1. Clear and objective conclusion

Response: Thanks for your favourable comments on our MS.

Reviewer 2 Report

Dear authors,

Thank you for the submission.

The topic is interesting as a stepping stone toward drug development of T. pisiformis in animals. There are few comments (all are in yellow highlighted colors) for further improvement of this manuscript.

Sincerely,
Stay safe!

Author Response

Reviewer 2

The topic is interesting as a steppingstone toward drug development of T. pisiformis in animals. There are few comments (all are in yellow highlighted colors) for further improvement of this manuscript.

Response: Thank you very much for your favourable comments and constructive suggestions on our MS. We have carefully revised our MS according to the comments in yellow highlighted colors.